# FiDeLiS: Faithful Reasoning in Large Language Models for Knowledge Graph Question Answering

**Yuan Sui**[1], **Yufei He**[1], **Nian Liu**[1], **Xiaoxin He**[1], **Kun Wang**[1,2], **Bryan Hooi**[1]*

[1]National University of Singapore
[2]University of Science and Technology of China
{yuansui, yufei.he, nianliu, xiaoxin, bhooi}@comp.nus.edu.sg,
wk520529@mail.ustc.edu.cn

## Abstract

Large language models (LLMs) are often challenged by generating erroneous or hallucinated responses, especially in complex reasoning tasks. Leveraging knowledge graphs (KGs) as external knowledge sources has emerged as a viable solution. However, existing KG-enhanced methods, either retrieval-based or agent-based, encounter difficulties in accurately retrieving knowledge and efficiently traversing KGs at scale. In this paper, we propose a unified framework, `FiDeLiS`, designed to improve the factuality of LLM responses by anchoring answers to verifiable reasoning steps retrieved from a KG. To achieve this, we leverage step-wise beam search with a deductive scoring function, allowing the LLM to validate each reasoning step and halt the search once the question is deducible. In addition, our `Path-rag` module pre-selects a smaller candidate set for each beam search step, reducing computational costs by narrowing the search space. Extensive experiments show that our training-free and efficient approach outperforms strong baselines, enhancing both factuality and interpretability. Code is released at https://anonymous.4open.science/r/FiDELIS-E7FC.

## 1 Introduction

Large language models (LLMs) have shown impressive reasoning capabilities in tackling complex tasks (Yu et al., 2024). However, the reasoning of LLMs is not always reliable and can be prone to generating outputs that are either inconsistent with real-world facts (Xu et al., 2024; Huang et al., 2025) or show flawed reasoning process (Li et al., 2024; Sui et al., 2024), which greatly undermine the reliability of LLMs in real-world applications (Kung et al., 2023; Zhang et al., 2024).

To address this issue, leveraging knowledge graphs (KGs) as external knowledge sources has emerged as a viable solution (Sun et al., 2023; Ma et al., 2024; Luo et al., 2024a). Unlike traditional retrieval-augmented generation (RAG) that relies on web pages or documents (Liu et al., 2024; Qian et al., 2024; Bayarri-Planas et al., 2024), KGs represent information in a structured and interconnected format, where each fact is stored as entities and relations. This format supports explicit, traceable reasoning processes (Pan et al., 2023) and facilitates multi-hop reasoning through graph traversal. Moreover, each fact in a KG can be traced back to its source (Sui et al., 2024; Agrawal et al., 2024), providing both context and original details, which further enhances the information authenticity and reliability of the reasoning processes.

Existing KG-enhanced LLM reasoning methods face notable challenges and can be roughly categorized into two primary approaches: retrieval-based and agent-based paradigms (Luo et al., 2024b). Retrieval-based methods (Wang et al., 2023; Luo et al., 2024a; Baek et al., 2023) retrieve relevant KG facts to support LLM reasoning by either prompting (Baek et al., 2023) or fine-tuning LLMs to learn the underlying structure of KG (Luo et al., 2024a;b). These methods often suffer from incomplete or imprecise information extraction due to a lack of contextual understanding or an inability to fully capture the graph structure (Luo et al., 2024b).Our error analysis of a strong retrieval-based method (i.e.,

---

* Corresponding author.

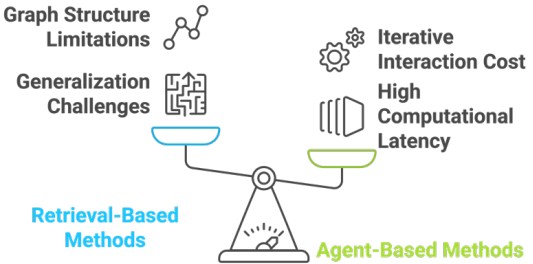

Figure 1: Challenges for existing KG-enhanced methods: How to balance faithfulness and efficiency?

(Luo et al., 2024a)) in §4.3 reveals that only 67% of the generated reasoning steps are valid, with 33% containing format errors or referencing non-existent KG facts. In contrast, agent-based methods (Sun et al., 2023; Ma et al., 2024) treat LLMs as interactive agents that explore KGs iteratively to construct reasoning paths and generate answers. While this approach by nature can enhance reasoning accuracy, it is computationally expensive, resulting in high latency and scalability limitations. As illustrated in Figure 1, balancing faithfulness and efficiency remains a critical challenge for existing KG-enhanced reasoning methods.

To this end, we propose **FiDeLiS**, a unified framework designed to improve the factual accuracy and reasoning efficiency of LLMs on KGQA task. FiDeLiS anchors LLM responses to verifiable reasoning steps derived from a KG by employing two core components: (1) `Deductive-Verification Beam Search (DVBS)` which systematically constructs and validates reasoning paths step-by-step, ensuring logical consistency and factual correctness (discussed in §3.2). This module also prevents premature reasoning termination and incorrect path extension to ensure the validity of the generated reasoning paths. (2) `Path-RAG`, a retrieval-augmented mechanism that pre-selects a constrained set of candidate entities and relations for each step to mitigate computational inefficiencies. It combines semantic similarity measures with graph-based connectivity analysis to optimize the search space, significantly reducing latency without sacrificing recall or accuracy (discussed in §3.1). Extensive experiments show that our method outperform strong baselines in both accuracy and efficiency, offering a scalable, training-free solution for KG-enhanced LLM reasoning. **Overall, our main contributions include:**

- We propose FiDeLiS, a unified framework designed to improve the factual accuracy of LLMs by grounding reasoning paths in structured KG efficiently.
- We enable efficient, verifiable reasoning by deductively validating reasoning steps and narrowing the search space with high-quality retrieval mechanism.
- FiDeLiS performs robustly across different experiments without the need for model fine-tuning, demonstrating adaptability and scalability with improved performance on multiple benchmarks.
- By anchoring responses in verifiable reasoning paths, FiDeLiS enhances interpretability, enabling users to verify and understand each reasoning step.

## 2  PRELIMINARY

**Notation.** To facilitate the demonstration of our method, we define the necessary notation below:

- *Definition 1.* A **reasoning step** is a pair $(r, e)$, where $r$ is the relation and $e$ is the corresponding entity.
- *Definition 2.* A **reasoning path** $\mathcal{P}$ is a pair $(s, \mathcal{T})$, where $s$ is the starting entity for the reasoning path, and $\mathcal{T}$ is a sequence of reasoning steps $\mathcal{T} = \{t_1, \dots, t_n\}$ and $t_k = (r_k, e_k)$ denotes the $k$-th reasoning step in the path and $n$ denotes the length of the path.
- *Definition 3.* The **next-hop candidates** given path $\mathcal{P}$, denoted $\mathcal{N}_1(e_n)$, is defined as the 1-hop neighborhood of $e_n$, the last node in the reasoning path $\mathcal{P}$.
- *Definition 4.* A reasoning path $\mathcal{P} = (s, \mathcal{T})$ is **valid** if every step $(r_k, e_k)$ corresponds to an actual triplet $(e_{k-1}, r_k, e_k)$ in the KG (with $e_0 = s$). For example, a valid reasoning path could be: $\mathcal{P} =$ Justin_Bieber $\xrightarrow{\text{people.person.son}}$ Jeremy_Bieber $\xrightarrow{\text{people.person.ex\_wife}}$ Erin_Wagner, which denotes that

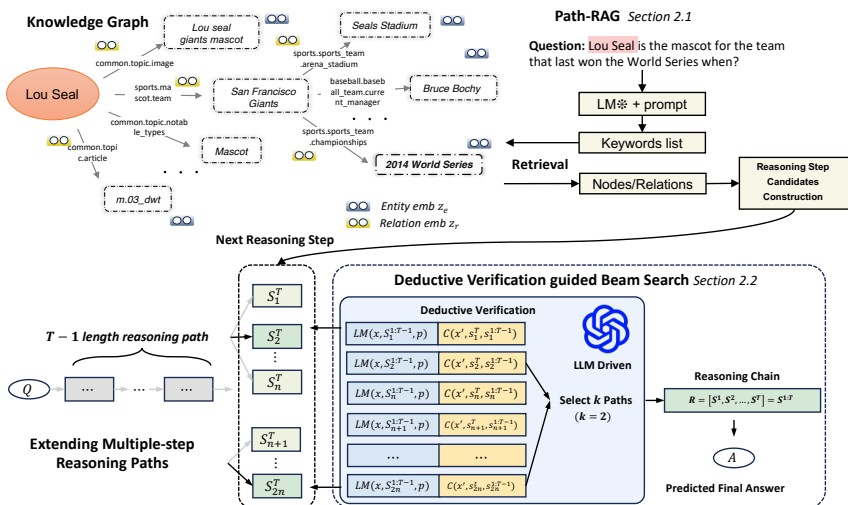

Figure 2: An illustration of FiDeLiS. **Top: The workflow of Path-RAG.** An LLM first extracts key terms and generates dense embeddings that feed into the Path-RAG module. Path-RAG rapidly retrieves relevant entities and relations from a pre-embedded KG and constructs candidate reasoning steps by combining semantic similarity with graph connectivity. **Bottom: The workflow of DVBS.** Then, the DVBS module uses an LLM-generated planning outline to guide a beam search that builds reasoning paths step-by-step over candidates constructed by Path-RAG, with deductive verification ensuring each step logically follows the previous steps and support the user question.

"Jeremy Bieber" is the father of "Justin Bieber" and "Erin Wagner" is the ex-wife of "Jeremy Bieber".

**Task definition.** In this work, we focus on the task of knowledge graph-based question answering (KGQA), a common reasoning task involving KGs. It is defined as: given a user query $q$ and a KG $\mathcal{G} = \{(e, r, e') \mid e, e' \in \mathcal{E}, r \in \mathcal{R}\}$, where $\mathcal{E}$ and $\mathcal{R}$ denote the set of entities and relations in KG, the task aims to design a function $f$ to predict answers $a \in \mathcal{A}_q$ conditioned on $q$ and $\mathcal{G}$. Following existing KG-enhanced LLMs methods (Sun et al., 2023; Ma et al., 2024), the function $f$ can be generally expressed as finding valid reasoning path(s) $\mathcal{P}$ on KGs that connects the entities mentioned in the query and the answer as: $P(a|q, \mathcal{G}) = \sum_{\mathcal{P}} P_\theta(a|q, \mathcal{P}) P_\phi(\mathcal{P}|q, \mathcal{G})$, where $P_\theta(a|q, \mathcal{P})$ denotes the probability of generating answer $a$ conditioned on $q$ and reasoning path(s) $\mathcal{P}$ by a function parameterized by $\theta$, and $P_\phi(\mathcal{P}|q, \mathcal{G})$ denotes the probability of discovering reasoning path(s) $\mathcal{P}$ by a function parameterized by $\phi$. As reasoning path $\mathcal{P}$ is defined as a sequence of reasoning steps, we factorize the reasoning path probability using the chain rule as Eq 1:

$$P(a|q, \mathcal{G}) = \sum_{\mathcal{P}} P_\theta(a|q, \mathcal{P}) \prod_{k=1}^{n} P_\phi(t_k|q, t_{<k}, \mathcal{G}) \tag{1}$$

To acquire valid reasoning paths, most prior studies follow the retrieval-based (Li et al., 2023; Luo et al., 2024a) or agent-based (Sun et al., 2023) paradigm. As indicated in Luo et al. (2024b), retrieval-based methods rely on precise additional retrievers, while agent-based methods are computationally intensive and lead to high latency. To address these issues, we propose our method, **FiDeLiS**, to enable both efficient and faithful reasoning over KGs.

## 3 METHOD

Motivated by the insight that integrating KGs with LLMs can mitigate hallucinations and enable verifiable reasoning, we propose FiDeLiS to improve the factuality of LLM responses by anchoring answers to verifiable reasoning steps retrieved from a KG. The overall framework of FiDeLiS is illustrated in Figure 2, which consists of two main components: (1) `Reasoning Path Retrieval-Augmented Generation` (Path-RAG, Algorithm 1) and (1) `Deductive-verification Beam Search` (DVBS, Algorithm 2).

Given a complex question $q$, we first use an LLM to extract key terms from $q$ and generate dense embeddings that capture the question's core concepts. These embeddings are input into Path-RAG module, which rapidly retrieves relevant entities and relations from a pre-embedded KG to select a smaller candidate sets for further beam search step, addressing the latency and computational burden of traditional agent-based methods. Path-RAG then constructs candidate reasoning steps by combining immediate semantic similarity with the structural connectivity of the graph, overcoming the dependence on highly precise retrievers in standard retrieval-based approaches. Next, the DVBS module employs an LLM-generated planning outline to guide a beam search that builds reasoning paths step-by-step. At each step, deductive verification checks that the accumulated reasoning steps logically supports the user question, ensuring the final reasoning path is both verifiable and accurate.

### 3.1 PATH-RAG: REASONING PATH RETRIEVAL-AUGMENTED GENERATION

Previous agent-based methods (Ma et al., 2024; Sun et al., 2023) treat LLMs as agents that iteratively interact with KGs to find reasoning paths and answer, which necessitate multiple rounds of interaction between agents and KGs and lead to high computational costs and latency. We instead propose a module, `Path-RAG` which iteratively pre-select a smaller candidate set to reduces the search space for exploring the potential reasoning paths from KGs. It consists of three steps and we detail the workflow as follows:

**Initialization.** We initiate the Path-RAG by encoding each entity $e_i \in \mathcal{E}$, and relation $r_i \in \mathcal{R}$ in the KG using a pre-trained language model (LM), which produces dense vectors $z(e^i) = \mathrm{LM}(e_i) \in \mathbb{R}^d$ and $z(r^i) = \mathrm{LM}(r_i) \in \mathbb{R}^d$, where $d$ denotes the embedding dimension. These embeddings are stored in a nearest neighbor structure to facilitate rapid similarity search.

**Keyword-Driven Retrieval.** We then populate a nearest neighbor index to retrieve relevant entities and relations for the user query. We first use an LLM to analysis the user query and generate exhaustive keywords/relation names that could be useful for finding the reasoning path to answer the query (See the prompt in §D.1). This step is designed to maximize coverage of potential reasoning steps, ensuring that no potential reasoning paths are overlooked during the retrieval process. The extracted keywords are then encoded using the same LM in initialization, yielding $z(K) = \mathrm{LM}(K) \in \mathbb{R}^d$. We subsequently compute the cosine similarity between $z(K)$ and the pre-stored embeddings, retrieving the top-$m$ entities and relations: $\mathcal{E}_m = \mathrm{argtopm}_{i \in |\mathcal{E}|} \cos\left(z(K), z(e_i)\right)$ and $\mathcal{R}_m = \mathrm{argtopm}_{i \in |\mathcal{R}|} \cos\left(z(K), z(r_i)\right)$.

**Reasoning Step Candidates Construction.** Next, we construct candidate reasoning steps defined in §2 using the retrieved candidate entities $\mathcal{E}_m$ and relations $\mathcal{R}_m$. To guide the selection of potential candidate, we propose a scoring function that combines semantic similarity with the KG's structural connectivity. First we define the base score function $S_0$ that captures only the semantic alignment of the candidate with the query as: $S_0((r,e)) = S_{\mathrm{rel}}(r) + S_{\mathrm{ent}}(e)$, where $S_{\mathrm{ent}}(e)$ and $S_{\mathrm{rel}}(r)$ represents the cosine similarity between the entity/relation to the query respectfully. To account for the KG's structural connectivity, *i.e.*, the potential for a candidate to lead to fruitful next steps, we incorporate information from the next-hop candidates and define the overall scoring function as Eq 2:

$$S((r,e)) = S_0((r,e)) + \alpha \max_{\forall (r_j, e_j) \in N(e)} S_0((r_j, e_j)) \tag{2}$$

Where $N(e)$ denotes the set of candidate relation-entity pairs reachable from entity $e$ within one hop in the KG. $\alpha$ is a hyper-parameter that balances the immediate semantic relevance (captured by $S_0((r,e))$) with the candidate's potential for future connectivity (captured by the maximum next-hop score). A higher $\alpha$ favors candidates with long-term benefits, even if they seem sub-optimal initially, while a lower $\alpha$ emphasizes immediate rewards, potentially overlooking future impacts. We verify the effectiveness of this new scoring function in Table 3 and append the tuning results of hyper-parameter $\alpha$ in Figure 5.

### 3.2 DEDUCTIVE-VERIFICATION BEAM SEARCH

The objective of `DVBS` is two-fold: (1) to provide a step-wise beam search for exploring verifiable reasoning paths from KG based on candidates constructed by Path-RAG (§3.1), and (2) to verify each reasoning step based on deductive reasoning (Ling et al., 2023) to ensure each step logically

follows the previous steps and supports the user query. Compared with existing methods like Sun et al. (2023) and Ma et al. (2024), while we both consider treat LLMs as agents that iteratively interact with KGs to find reasoning paths and answers, our method leverage deductive reasoning to ensure each reasoning steps are logically connected and only halt the search if the question can be deduced based on the reasoning paths. Based on our robustness analysis in §4.3, DVBS demonstrate higher ratio of valid reasoning paths and can prevent issues of either premature stopping (Huang et al., 2017) or excessive continuation of reasoning path extension. The DVBS consists of three steps and we detail the workflow as follows:

**Plan Generation.** Inspired by the recent works regarding planning capabilities of LLMs (Zhang et al., 2023; Kagaya et al., 2024), we prompt an LLM to generate the planning steps for answering the user query, denoted as $w$. This step is designed to provide more hints for subsequent LLM decision making process. Even this step is more like an engineering trick, we find that it may unlock some of the capabilities of LLM to do "higher-order" thinking. By including more hints in the prompt, the LLM tends to make more accurate and deterministic decisions during beam search, thus improving the quality of the traversed reasoning paths. (See Table 2 for ablation analysis).

**Beam Search.** We then construct the multi-step reasoning paths by iteratively extending partial paths using a beam search strategy. At each time step $t$, we use an LLM as agent to select one reasoning step $s^t$ from a candidates set $\mathcal{S}^t$ (§3.1) conditioned on (1) the likelihood of each candidate $s_i \in \mathcal{S}^t$, (2) the user query $q$, (3) the history of previous steps $s^{1:t-1}$, and (4) planning context $w$ (§3.2), denoted as $\mathrm{LM}(s^t|q, s^{1:t-1}, w, \mathcal{S}^t)$. Instead of exploring every possibility, we retain only the top-$k$ scoring paths from the previous beam $\mathcal{H}_{t-1}$ and extend them by appending candidate steps. The overall process can be expressed as Eq 3:

$$\mathcal{H}_t = \mathrm{Top}_k \{ h \oplus \mathrm{LM}(s^t|q, s^{1:t-1}, w, \mathcal{S}^t) : h \in \mathcal{H}_{t-1} \} \tag{3}$$

where $\oplus$ denotes the concatenation of the current path $h$ with the selected candidate step $s^t$. The beam search strategy enable efficiently navigate the vast space of potential reasoning paths while concentrating on the most promising ones.

While beam search, by its nature, can incur high computational costs and latency due to multiple rounds of LLM interactions. Our retrieval module Path-RAG mitigate this issue by constraining candidate set $\mathcal{S}^t$ at each time step $t$ to a narrow, high-quality subset rather than requiring the LLM to consider all available options. This targeted retrieval not only reduces the number of candidates to evaluate at each step but also increases the likelihood of selecting relevant reasoning steps, thereby enabling efficient traversal of KGs at scale. Find more discussion regarding efficiency of FiDeLiS in §4.4 and Appendix §B.

**Deductive Verification.** To ensure that each reasoning step logically follows from its predecessors and adequately supports the original query, we leverage the deductive reasoning capabilities of LLMs as a verification criterion (Ling et al., 2023) for the beam search process. We first convert the user query $q$ into a clear declarative statement $q'$, which encapsulates its logical intent and allows the LLM to operate on a well-defined logical target (See the concrete example in §D.6). Next, during the beam search, candidate reasoning step $s^t$ are appended to the history $s^{1:t-1}$ to form potential reasoning paths. For each candidate, we then invoke two deductive verification checks, $C_{\mathrm{global}}$ and $C_{\mathrm{local}}$ (the prompts are given in §D.2). Only those candidates that pass local verification, indicating that the new step maintains logical consistency with the established context, are retained in the beam search process. Once the candidates pass both verification indicate that the user query $q$ can be deduced based on the retained reasoning paths $s^{1:t}$ and the beam search progress should be halted.

> **Global Verification:** $C_{\mathrm{global}}(s^{1:t-1}, s^t)$ returns 1 if $(s^t \wedge s^{1:t-1}) \models q'$, and 0 otherwise.

> **Local Verification:** $C_{\mathrm{local}}(s^{1:t-1}, s^t)$ returns 1 if $s^t$ logically follows from $s^{1:t-1}$, and 0 otherwise.

By integrating this verification into the beam search offers several benefits: it (1) enhances the robustness and validity of the final answer by enforcing logical coherence at every step, (2) reduces

computational overhead by pruning unpromising paths early, and (3) mitigates risks such as premature termination or excessive extension of the reasoning process. We provide a concrete example of the deductive verification process in §D.6 and the complete DVBS algorithm in Algorithm 2.

## 4 EXPERIMENTS

In this section, we focus on verifying FiDeLiS from four perspectives as follows: (1) comparison results with other baselines over KGQA; (2) ablation study; (3) robustness analysis and (4) efficiency analysis. We provide **all the experiment settings** in Appendix A due to page constraints. The prompts for plan generation, beam search and deductive verification can be found in §D.

### 4.1 MAIN RESULTS

In Table 1, we compare the performance of different methods with various backbend LLMs across three datasets. We found that LLM + KG approaches generally outperform LLM-only methods (Zero-shot, Few-shot, and CoT) by a wide margin, indicating the significant benefit of incorporating KGs into LLM reasoning. In the LLM + KG category, FiDeLiS stands out as the best-performing method across all datasets, particularly when paired with GPT-4-turbo. For example, on WebQSP, FiDeLiS achieves 84.39% Hits@1 and 78.32% F1, surpassing ToG (81.84% Hits@1, 75.97% F1) and RoG (83.15% Hits@1, 69.81% F1). This improvement is consistent across other datasets, and even compared with some finetuning methods like DeCAF and RoG, FiDeLiS as a training-free method still demonstrate better performance. The consistent performance of FiDeLiS highlights its effective use of both the KG and LLM, as well as its optimization of hyper-parameters like beam width and depth. Overall, the results illustrate that FiDeLiS, leveraging advanced LLMs like GPT-4-turbo and KG-based reasoning, sets a new standard for performance in KG-related tasks.

| Backend Models | Methods | WebQSP | | CWQ | | CR-LT |
|---|---|---|---|---|---|---|
| | | Hits@1 (%) | F1 (%) | Hits@1 (%) | F1 (%) | Acc (%) |
| Prompting - LLM Only `gpt-3.5-turbo` | Zero-shot | 54.37 | 52.31 | 34.87 | 28.32 | 32.74 |
| | Few-shot | 56.33 | 53.12 | 38.52 | 33.87 | 36.61 |
| | CoT | 57.42 | 54.72 | 43.21 | 35.85 | 37.42 |
| Prompting - LLM Only `gpt-4-turbo` | IO | 62.32 | 59.71 | 42.71 | 37.93 | 37.74 |
| | Few-shot | 68.65 | 62.71 | 51.52 | 43.70 | 43.61 |
| | CoT | 72.11 | 65.37 | 53.51 | 44.76 | 45.42 |
| Finetuning - LLM + KG | NSM (He et al., 2021) | 74.31 | - | 53.92 | - | - |
| | CBR-KBQA (Das et al., 2021) | - | - | 67.14 | - | - |
| | DeCAF (Yu et al., 2023) | 82.1 | - | 70.42 | - | - |
| | KD-CoT (Wang et al., 2023) | 73.7 | 50.2 | 50.5 | - | - |
| | RoG (Luo et al., 2024a) | 83.15 | 69.81 | 61.39 | 56.17 | 60.32 |
| Prompting - LLM + KG `gpt-3.5-turbo` | ToG (Sun et al., 2023) | 75.13 | 72.32 | 57.59 | 56.96 | 62.48 |
| | KAPING (Baek et al., 2023) | 72.42 | 65.12 | 53.42 | 50.32 | - |
| | FiDeLiS | 79.32 | 76.78 | 63.12 | 61.78 | 67.34 |
| Prompting - LLM + KG `gpt-4-turbo` | ToG (Sun et al., 2023) | 81.84 | 75.97 | 68.51 | 60.20 | 67.24 |
| | FiDeLiS | **84.39** | **78.32** | **71.47** | **64.32** | **72.12** |

Table 1: Comparison of FiDeLiS with baseline methods and different backbone LLMs. We replicate the outcomes of ToG and RoG, and retrieve other baseline results directly from the original paper. We utilize 5 demonstrations as our default setting for FiDeLiS, ToG, Few-shot, and CoT. The experiment results of open-source models can be found in Table 11.

### 4.2 ABLATION STUDY

Table 2 demonstrates the ablation study of FiDeLiS using the `gpt-3.5-turbo-0125` model, highlighting the contributions of individual components (Path-RAG and DVBS) to overall performance. We conduct the ablation of the Path-RAG by replacing it with either a vanilla retriever or ToG (Sun et al., 2023) as retriever. We find that using ToG shows slight improvements over the vanilla retriever but remains below using Path-RAG. Ablating DVBS components also leads to performance declines, particularly when beam search is removed, causing Hits@1 on WebQSP to drop sharply to 60.35%. The deductive verifier and last-step reasoning show moderate but noticeable impacts on performance. The effects are less pronounced on CR-LT, suggesting it is more tolerant of simpler methods. Overall, the results confirm the critical roles of Path-RAG and DVBS, especially beam search, in ensuring robust and accurate reasoning across domains.

| Ablation Setting | Components | WebQSP | CWQ | CR-LT |
|---|---|---|---|---|
| | | Hits@1 (%) | Hits@1 (%) | Acc (%) |
| No ablation | FiDeLiS | 79.32 | 63.12 | 67.34 |
| w/o Path-RAG | using vanilla retriever | 72.35 | 57.11 | **59.78** |
| | using ToG | 75.11 | 59.47 | 63.47 |
| w/o DVBS | w/o last step reasoning | 75.68 | 59.45 | 63.72 |
| | w/o planning | 76.23 | 60.14 | 64.13 |
| | w/o beam-search | **60.35** | **49.78** | 61.87 |
| | w/o deductive-verifier | 74.13 | 57.23 | 63.89 |

Table 2: Ablation Studies of FiDeLiS using model `gpt-3.5-turbo-0125`. $\Delta$ refers to the performance gap between each component and the entire method.

| Methods | Backbones | WebQSP | CWQ | CR-LT |
|---|---|---|---|---|
| | | Hits@1 (%) | Hits@1 (%) | Acc (%) |
| Vanilla Retriever | w/ BM25 | 58.31 | 48.39 | 50.73 |
| | w/ SentenceBert | 62.74 | 50.14 | 51.80 |
| | w/ E5 | 68.42 | 52.84 | 54.31 |
| | w/ Openai-Emb* | **72.35** | **57.11** | **59.78** |
| Path-RAG | w/ BM25 | 70.34 | 56.11 | 58.77 |
| | w/ SentenceBert | 73.45 | 58.41 | 60.45 |
| | w/ E5 | 77.93 | 62.74 | 65.23 |
| | w/ Openai-Emb* | **79.32** | **63.12** | **67.34** |

Table 3: Performance of FiDeLiS with various embedding methods. * refers to `text-embedding-3-small` from OpenAI. We detail the tested embedding methods in §A.3.

## 4.3 ROBUSTNESS ANALYSIS

**Robustness of Path-RAG.** Table 3 presents the performance of FiDeLiS compared to a vanilla retriever with different embedding methods. The results consistently show that FiDeLiS outperforms the vanilla retriever irrespective of the underlying embedding strategy. For instance, with Openai-Emb*, the vanilla retriever achieves 72.35% on WebQSP, whereas Path-RAG reaches 79.32%, indicating a notable improvement. Similar performance gains are observed with the other embeddings. These improvements suggest that integrating graph connections can enhance retrieval effectiveness by providing more informative and contextually relevant information, thereby bolstering the overall robustness and accuracy of the method.

**Effectiveness of Path-RAG.** We verify the effectiveness of the retrieval module `Path-RAG` with two baselines: (1) a vanilla retriever and (2) KAPING (Baek et al., 2023) method. The vanilla retriever concatenates each entity with its relation to form a reasoning step and selects candidates based on cosine similarity with the query embeddings. In contrast, KAPING (Baek et al., 2023) converts each triple into text and retrieves the top-$K$ similar triples based on semantic similarity. We quantify the retrieval performance using the coverage ratio (CR), defined as the percentage of the ground-truth reasoning steps being retrieved throughout the reasoning path extension (i.e., $\text{CR} = \frac{N_{\text{retrieved}} \cap N_{\text{ground-truth}}}{N_{\text{ground-truth}}}$). Table 4 illustrate the experimental setup and corresponding results. We find that compared with the baselines, our Path-RAG achieves a higher CR value and aligns better with the ground-truth paths. It demonstrates superior ability to capture connections that simpler retrieval models may overlook. This advantage is critical for guiding subsequent LLM processing toward relevant information, ultimately yielding more accurate and coherent answers.

| Method | Depth = 1 | Depth = 2 | Depth > 3 |
|---|---|---|---|
| Vanilla Retriever | 59.34 | 52.17 | 47.31 |
| KAPING (Baek et al., 2023) | 65.72 | 60.41 | 53.11 |
| Path-RAG w/ keywords | 72.61 | 69.38 | 62.78 |
| Path-RAG w/o keywords | 68.78 ($\downarrow$ 3.83) | 65.27 ($\downarrow$ 4.11) | 57.13 ($\downarrow$ 5.65) |

Table 4: Analysis of the CR of reasoning paths over CWQ.

**Path Error Analysis.** To verify the faithfulness of our step-wise method, we conduct an error analysis regarding the whole reasoning path generation using RoG (Luo et al., 2024a). We quantify the validity of reasoning path using validity ratio (VR), which is defined as the ratio of reasoning

| Methods | WebQSP (hits@1) | CWQ (hits@1) |
|---|---|---|
| Deductive Verification | 79.32 | 63.12 |
| Adequacy Verification (used in ToG) | 74.13 | 57.23 |
| Logit-based Scoring | 73.47 | 54.78 |

Table 5: Analysis of different verification methods.

| Dataset | Method | Hits@1 | Runtime | Token | # |
|---|---|---|---|---|---|
| WebQSP | FiDeLiS (ours) | 79.32 | 43.83 | 2,452 | 10.7 |
| | w/o Path-RAG using vanilla retriever | 72.35 | 48.37 | 2,873 | 10.7 |
| | w/o Path-RAG using ToG | 75.11 | 74.26 | 6,437 | 10.7 |
| | FiDeLiS (ours) - GPT-4o | **81.17** | 37.82 | 2,452 | 10.7 |
| | FiDeLiS (ours) - GPT-4o-mini | 76.48 | **24.31** | 2,452 | 10.7 |
| CWQ | FiDeLiS (ours) | 63.12 | 74.59 | 2,741 | 15.2 |
| | w/o Path-RAG using vanilla retriever | 57.11 | 78.41 | 3,093 | 15.2 |
| | w/o Path-RAG using ToG | 59.47 | 132.59 | 5,372 | 15.2 |
| | FiDeLiS (ours) - GPT-4o | **65.33** | 50.12 | 2,741 | 15.2 |
| | FiDeLiS (ours) - GPT-4o-mini | 58.34 | **42.54** | 2,741 | 15.2 |

Table 7: Runtime efficiency of FiDeLiS per question.

steps that existed in the KG to the total number of the reasoning steps in the output reasoning path (i.e., $VR = \frac{N_{\text{valid-steps}}}{N_{\text{all-steps}}}$). As shown in Figure 3, only 67% of generated reasoning steps are valid, while the remaining 33% of reasoning steps either have a format error or do not exist in the KG. This illustrates that the reasoning steps generated offer few guarantees about feasibility especially when multiple consecutive steps are combined into a reasoning path. While our method leverage step-wise verification to ensure that each of the reasoning step exist in the KG and logically connected.

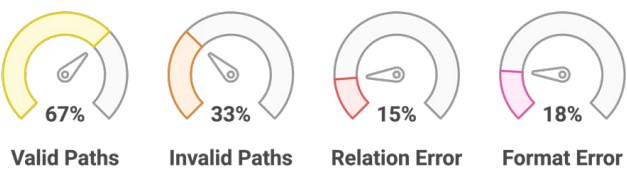

Figure 3: Analysis of reasoning errors in RoG (Luo et al., 2024a) over WebQSP.

**Effectiveness of Deductive-Verification.** To verify the effectiveness of deductive-verification mentioned in §3.2, we calculate the average depths of the generated reasoning paths as shown in Table 6. We find that by considering deductive verification, it consistently shows shorter and closer reasoning depths to ground-truth across all datasets compared to baseline. This implies that FiDeLiS may offer more precise termination signals and potentially more accurate reasoning paths. We also compare deductive-verification methods with other baselines in Table 5, like logit-based scoring that assign softmax probability scores to determine the endpoint of beam search process, and adequacy verification used in ToG (Sun et al., 2023). Experiments show higher accuracy with deductive verification compared to adequacy verification and logit-based scoring, demonstrating its effectiveness in enhancing reasoning accuracy.

This finding is further supported by a case study regarding a complex question of Iran's government system, which blends elements of religion and democracy as shown in Table 8. While baseline methods such as GPT + CoT and RoG predominantly identified Iran as an "Islamic Republic" and ToG produce mixed responses, our approach—enhanced by deductive verification—delivers a reasoning path that is both concise and context-aware. The proposed verification mechanism not only streamlines the reasoning process but also ensures comprehensive coverage of grounded answers, demonstrating FiDeLiS's strength in handling intricate questions.

| Method | WebQSP | CWQ | CR-LT |
|---|---|---|---|
| GT | 2.3 | 3.2 | 4.7 |
| ToG | 3.1 | 4.1 | 5.2 |
| FiDeLiS | 2.4 | 2.8 | 4.6 |

Table 6: Average depths of the returned reasoning paths. GT refers to ground-truth reasoning paths.

| Question | What form of government is in the country that uses the Iranian Rail and was established in 1979? |
|---|---|
| Answer | 'Parliamentary system', 'Presidential system', 'Islamic republic', 'Unitary state', 'Theocracy' |
| GPT + CoT | **Think step by step.** First, identify the country: the country that uses the Iranian Rail is Iran. Then, retrieve the historical Context of 1979: In 1979, Iran underwent a significant transformation with the Iranian Revolution, which led to the overthrow of the Pahlavi monarchy. Third, following the revolution, the country adopted a new form of government based on Islamic law. The result was the establishment of the Islamic Republic of Iran, which combines elements of a theocracy and a democracy, where religious leaders have significant influence over the state governance. So, the government form is the Islamic Republic of Iran, established in 1979 following the Iranian Revolution. |
| RoG | location.country.form_of_government, location.location.containedby location.country.form_of_government' 
 location.location.containedby → government.form_of_government.countries 
 The results are Islamic republic. |
| ToG | **Path-1**: Iranian rial → finance.currency.countries_used → Iran → location.country.form_of_government → Islamic republic → government.form_of_government.countries → Iran 
 **Path-2**: Iranian rial → finance.currency.countries_used → Iran → location.country.form_of_government → Theocracy → government.form_of_government.countries → Iran 
 **Path-3**: Iranian rial → finance.currency.countries_used → Iran → location.country.form_of_government → Unitary state → government.form_of_government.countries → Iran 
 Based on the reasoning paths, the result is Iran. |
| FiDeLiS | **Path-1**: Iranian rial → finance.currency.countries_used → Iran → location.country.form_of_government → Islamic republic 
 **Path-2**: Iranian rial → finance.currency.countries_used → Iran → location.country.form_of_government → Theocracy 
 **Path-3**: Iranian rial → finance.currency.countries_used → Iran → location.country.form_of_government → Unitary state 
 Based on the reasoning paths, the results are Theocracy, Unitary state, Islamic republic. |

Table 8: Case study of FiDeLiS. We highlight the wrong answers with red color, and correct answers with blue color.

## 4.4 EFFICIENCY ANALYSIS

To investigate the runtime efficiency and cost efficiency of FiDeLiS, we present a comparison regarding the average runtime, average token usage, average times of LLM calling per question in Table 7. We find that (1) our method shows superior efficiency compared to the ToG (which is also training-free), by reducing approximately 1.7x runtime costs. (2) Path-Rag component is critical in enhancing both the accuracy and efficiency of the model. Its ability to constrain potential path candidates effectively reduces unnecessary computational overhead, leading to quicker and more accurate results. To address concern regarding our method's potential application in real-time scenarios, we also test our method using faster and more advanced LLMs. Table 7 shows that our method could be further accelerated with newer, faster models like GPT-4o or GPT-4-mini. The potential of the ongoing advancements in LLMs are expected to further enhance the scalability and efficiency of FiDeLiS, making it a practical development in challenging environments. More detailed analysis of bottleneck of computation of FiDeLiS can be further found in Appendix B.

## 5 RELATED WORK

**LLM Reasoning & Role of KGs.** Large language models (LLMs) demonstrate impressive capabilities in reasoning tasks but often generate hallucinated or factually incorrect outputs, particularly in complex, multi-step scenarios (Huang et al., 2025; Li et al., 2024). This unreliability reduces their effectiveness in knowledge-intensive applications. Knowledge graphs (KGs) have emerged as a solution by offering structured, verifiable data that supports transparent and multi-hop reasoning (Sui et al., 2024). Unlike document-based retrieval-augmented generation approaches, KGs provide direct access to relational facts, enhancing both interpretability and traceability (Chen et al., 2024).

**KG-enhanced LLM Reasoning.** KG-enhanced reasoning methods are generally categorized into retrieval-based and agent-based models. Retrieval-based approaches, such as DeCAF (Yu et al., 2023), rely on text-based retrieval to select relevant information from KGs and jointly generate answers and logical forms, but their performance can degrade without precise retrieval mechanisms. In contrast, agent-based models, like ToG (Sun et al., 2023), iteratively explore reasoning paths but suffer from high computational overhead. To address these limitations, recent methods like RoG (Luo et al., 2024a) and GCR (Luo et al., 2024b) have sought to integrate KG structure into LLM training or decoding to improve reasoning fidelity and explanation generation. To improve the faithfulness of the LLM reasoning, KD-CoT (Wang et al., 2023) verifies sub-reasoning steps through external KGs to prevent errors during inference, while NSM (He et al., 2021) employs a teacher-student architecture to learn intermediate supervision signals that guide reasoning.

## 6 CONCLUSION

This paper proposes a retrieval-exploration interactive method specifically designed to enhance intermediate steps of LLM reasoning grounded by KGs. The Path-RAG module and the use of deductive reasoning as a calibration tool effectively guide the reasoning process, leading to more accurate knowledge retrieval and prevention of misleading reasoning chains. Extensive experiments

demonstrate that our method, being training-free, not only reduces computational costs but also offers superior generality. We believe this study will significantly benefit the integration of LLMs and KGs, or serve as an auxiliary tool to enhance the interpretability and factual reliability of LLM outputs.

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

# A  EXPERIMENT DETAILS

## A.1  BASELINES

We consider the following methods including training-free (highlighted with *) and training-based methods as baselines:

- NSM He et al. (2021) propose a teacher-student approach for KGQA task, where the student network aims to find the correct answer to the query, while the teacher network tries to learn intermediate supervision signals for improving the reasoning capacity of the student network.
- KD-CoT Wang et al. (2023) propose to verify the sub-reasoning process of LLMs through the external KGs to facilitate faithful reasoning.
- DeCAF Yu et al. (2023) use a text-based retrieval instead of entity linking to select question-related information from the KG, and generate logical forms and direct answers respectively. They combine the logical-form-executed answers and directly-generated answers to obtain the final output.
- KAPING* Baek et al. (2023) proposes a zero-shot knowledge-augmented prompting method. It first retrieves triples related to the question from the graph, then prepends them to the input question in the form of a prompt, which is then forwarded to LLMs to generate the answer.
- ToG* Sun et al. (2023): conduct the reasoning on KGs by iteratively exploring multiple potential reasoning paths and concludes the final answer by aggregating the evidence from retrieved reasoning paths.
- RoG Luo et al. (2024a): incorporate the underling structure of KGs into LLMs throught pre-training and fine-tuning to generate the reasoning path and explanation.
- GCR Luo et al. (2024b) propose to integrate KG structure into the LLM decoding process to conduct graph-constrained reasoning.

## A.2  DATASETS & METRICS

We consider three KGQA benchmark: WebQuestionSP (WebQSP) Yih et al. (2016), Complex WebQuestions (CWQ) Talmor & Berant (2018) and CR-LT-KGQA Guo et al. (2024) in this work. We follow previous work Luo et al. (2024a) to use the same training and testing splits for fair comparison over WebQSP and CWQ. The questions from both WebQSP and CWQ can be reasoned using Freebase KGs[1]. To address the bias in WebQSP and CWQ, which predominantly feature popular entities and there is a likelihood that their data might have been incorporated into the pre-training corpora of LLMs, we further test our method on CR-LT-KGQA (discussed in §A.2). We use the complete dataset from CR-LT-KGQA in our experiments, as it comprises only 200 samples. Each of the question can be reasoned based on the Wikidata[2]. The statistics of the datasets are given in Table 10 and Table 9. To streamline the KGs, we follow RoG Luo et al. (2024a) and utilize a subgraph of Freebase by extracting all triples that fall within the maximum reasoning hops from the question entities in WebQSP and CWQ. Similarly, we construct the corresponding sub-graphs of Wikidata for CR-LT-KGQA. We assess the performance of the methods by analyzing the F1 and Hits@1 metrics for CWQ and WebQSP, and by evaluating the accuracy for CR-LT-KGQA. The statistics of the datasets can be found in Table 9 and Table 10.

| Dataset | 1 hop | 2 hop | $\geq$ 3 hop |
|---------|-------|-------|--------|
| WebQSP | 65.49 % | 34.51% | 0.00% |
| CWQ | 40.91 % | 38.34% | 20.75% |
| CR-LT | 5.31 % | 43.22% | 51.57% |

Table 9: Statistics of the question hops in WebQSP, CWQ and CR-LT-KGQA.

**Motivation of CR-LT-KGQA.** The motivation for evaluating over CR-LT-KGQA is that the majority of existing KGQA datasets, including WebQSP and CWQ, predominantly feature popular entities. These entities are well-represented in the training corpora of LLMs, allowing to often

---

[1] https://github.com/microsoft/FastRDFStore
[2] https://www.wikidata.org/wiki/Wikidata:Main_Page

| Dataset | #Ans = 1 | 2 ≥ #Ans ≤ 4 | 5 ≥ #Ans ≤ 9 | #Ans ≥ 10 |
|---------|----------|--------------|--------------|-----------|
| WebQSP | 51.2% | 27.4% | 8.3% | 12.1% |
| CWQ | 70.6% | 19.4% | 6% | 4% |

Table 10: Statistics of the number of answers for questions in WebQSP and CWQ.

generate correct answers based on their internal knowledge, potentially without external KGs. Moreover, since WebQSP and CWQ have been available for several years, there is a likelihood that their data might have been incorporated into the pre-training corpora of LLMs, further reducing the need for external KGs during question-answering. **To this end**, we utilize the CR-LT-KGQA benchmark, which features queries specifically crafted to target obscure and long-tail entities. Figure 4 illustrates the distribution of entity frequency and popularity in CR-LT, underscoring the inherent challenges of these queries. In such scenarios, knowledge graphs are indispensable as they offer a reliable, verifiable source of information, particularly for entities that are poorly represented in the training data of large language models. By testing our methods on CR-LT-KGQA, we investigate the extent to which integrating KGs can bolster LLM performance in less common knowledge domains, where their effectiveness typically declines. This evaluation not only demonstrates the potential synergy between LLMs and KGs but also clarifies the critical role that KGs continue to play in supporting LLMs across diverse query scenarios.

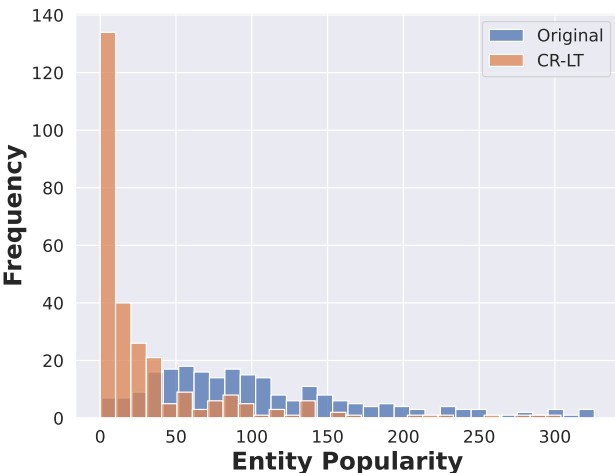

Figure 4: Distribution of CR-LT-KGQA dataset.

### A.3 BACKBONE LLMS & EMBEDDING METHODS

**Backbone LLMs.** We assess our approach on closed- and open-source LLMs. We consider closed-source models like GPT-4-turbo (between Feb, 2024 to July, 2024), GPT-3.5-turbo (between Feb, 2024 to July, 2024), GPT-4o, GPT-4o-mini (between Nov, 2024 to Jan 2025) from OpenAI, and open-sourced models like meta-llama-2-13B from Meta and mixtral-7B from Mixtral AI. The experiment results of open-source models can be found in Table 11. We set all the inference configs using temperature $T = 0.3$ and $p = 1.0$.

**Embedding Methods.** We assess the robustness of the retrieval module Path-RAG on different embedding models. We consider probabilistic ranking function like BM25[3], dense retrieval using smaller language models like SentenceBERT[4] and E5[5], and more advanced embedding model like text-embedding-3-small from OpenAI[6].

---

[3] https://en.wikipedia.org/wiki/Okapi_BM25
[4] https://sbert.net
[5] https://huggingface.co/intfloat/e5-large
[6] https://platform.openai.com/docs/guides/embeddings

| Backend Models | WebQSP | | CWQ | | CR-LT |
| --- | --- | --- | --- | --- | --- |
| | Hits@1 (%) | F1 (%) | Hits@1 (%) | F1 (%) | Acc (%) |
| Llama-2-13B | 72.34 | 69.78 | 58.41 | 54.78 | 60.87 |
| Mistral-7B | 74.11 | 70.23 | 60.71 | 56.87 | 63.12 |

Table 11: Performance over Open-sourced LLMs.

## A.4 IMPLEMENTATION DETAILS

We set the default beam width as $4$ and depth as $4$ without specific annotation. We set the $\alpha$ in Eq 2 as $0.3$ to ensure reproducibility. For hyper-parameter tuning regarding $\alpha$ for Eq 2 and beam search width and length, we conduct experiments as shown in Figure 5 and Figure 6.

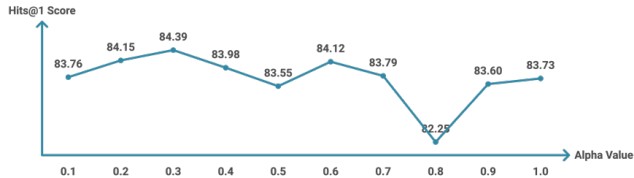

Figure 5: Parameter tuning for $\alpha$ for scoring function over WebQSP

**Analysis of Beam Search.** We investigate the effect of hyper-parameters like beam width and depth in beam search, as illustrated in Figure 6. By varying the width and depth from 1 to 4, we observe that overall performance improves as both parameters increase, peaking when the search depth exceeds 3 for the WebQSP and CWQ datasets. However, beyond a depth of 3, performance begins to decline, likely because only a small fraction of questions in these datasets require reasoning at greater depths. In contrast, increasing the beam width consistently enhances performance, highlighting the benefits of broader exploration in search.

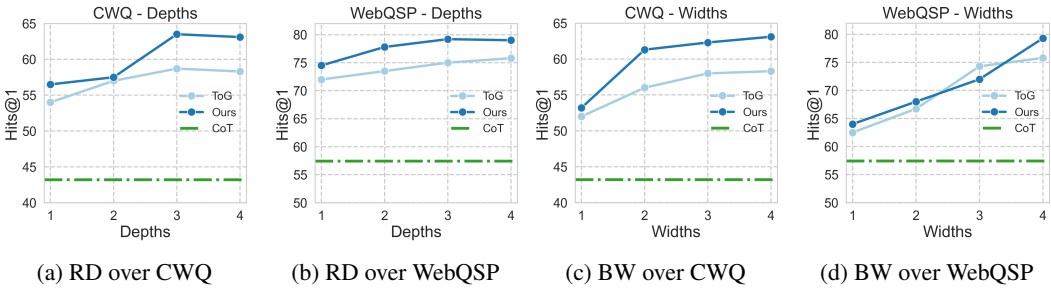

(a) RD over CWQ     (b) RD over WebQSP     (c) BW over CWQ     (d) BW over WebQSP

Figure 6: Analysis of various beam search width (BW) and reasoning depth (RD).

## A.5 ROBUSTNESS ANALYSIS ACROSS DIFFERENT DOMAINS AND KGS

KGs vary in structure and domain-specific characteristics, so consistent performance across both general and specialized KGs can reflect a method's adaptability to diverse real-world applications. To this end, we conduct robustness analysis of FiDeLiS across different domains and KGs to verify the generalizability. To perform this analysis, we introduced a new dataset, MedQA-USMIE, sourced from MedQA Jin et al. (2020), which is designed to require domain-specific biomedical and clinical knowledge. The dataset is a 4-way multiple-choice question-answering task, and we extracted 300 examples from its test set for evaluation. The corresponding biomedical KG is based on Disease Database and DrugBank Zhang et al. (2022). The results, presented in Table 2, indicate that our method exhibits consistent robustness across different types of KGs. Our scoring function, enhanced by incorporating next-hop neighbor information $S_r^i + S_e^i + \alpha \max_{\forall j \in N_i}(S_r^j + S_e^j)$, achieves higher performance gains in both WebQSP and MedQA-USMIE, particularly improving accuracy in the specialized biomedical domain. These findings validate that our method can effectively handle

| Method | WebQSP | MedQA-USMIE |
|---|---|---|
| ToG | 81.84 | 42.37 |
| Path-RAG w/ $S_r^i + S_e^i$ | 83.15 | 44.31 |
| Path-RAG w/ $S_r^i + S_e^i + \alpha \max_{\forall j \in N_i}(S_r^j + S_e^j)$ | **84.39** | **46.45** |

Table 12: Robustness analysis of our method across different domains

the challenges posed by both general and domain-specific knowledge graphs, indicating strong adaptability and robustness.

## B  BOTTLENECK OF BEAM SEARCH EFFICIENCY

The bottleneck of computation is the beam search process, which contributes to $N * D$ times LLM calling, where $D$ is the depth (or equivalently length) of the reasoning path, and $N$ is the width of the beam-search (how many paths are remained in the pool in each iteration). Specifically, we need to call $ND + D + C$ times LLM for each sample question, where $C$ is a constant (equals to 1 if there is no error occurs when calling the API). Sun et al. (2023) indicate that the computational efficiency can be alleviated by replacing LLMs with small models such as BM25 and Sentence-BERT for the beam search decision since the small models are much faster than LLM calling. In this way, we can reduce the number of LLM calling from $ND + D + C$ to $D + C$. However, this may sacrifices the accuracy due to the weaker scoring model in decision making Sun et al. (2023).

We noted that $ND + D + C$ is the maximal computational complexity. In most cases, FiDeLiS does not need $ND + D + C$ LLM calls for a question because the whole reasoning process might be early stopped before the maximum reasoning depth $D$ is reached if LLM determines the query can be deductive reasoning by the current retrieved reasoning paths. As an illustration, Table 7 shows the average numbers of LLM calls per question needed by FiDeLiS on different datasets. It can be seen that in three KGQA datasets, the average numbers of LLM calls (ranging from ) are smaller than 21, which is the theoretical maximum number of LLM calls calculated from $ND + D + C$ when $N = 4$ and $D = 4$. We can also see that this average number gets even smaller for dataset covering a lot of single-hop reasoning questions, such as WebQSP.

## C  LIMITATIONS

Our work demonstrates a promising advancement by integrating KGs with LLMs to reduce hallucinations and promote deep, faithful reasoning through deductive verification. However, the method exhibits certain limitations. Its reliance on external KGs means that the overall effectiveness is contingent on the quality and comprehensiveness of these resources, and challenges may arise when encountering incomplete, inconsistent or outdated information. Despite these limitations, the open-KGs like Wikidata and DBpedia used in our study are of high quality, benefiting from years of updates by an extensive community. For domain-specific KGs, although there may currently be gaps in quality, we are optimistic about future enhancements. Given the significant societal impact and the noticeable boosts in LLM performance facilitated by KGs, it is likely that community efforts will continue to refine and expand these resources.

## D   PROMPT LIST

In this section, we show all the prompts that need to be used in the main experiments. The `In-Context Few-shot` refers to the few-shot examples we used for in-context learning.

### D.1   PLAN-AND-SOLVE

You are a helpful assistant designed to output JSON that aids in navigating a knowledge graph to answer a provided question. The response should include the following keys:

(1) 'keywords': an exhaustive list of keywords or relation names that you would use to find the reasoning path from the knowledge graph to answer the question. Aim for maximum coverage to ensure no potential reasoning paths will be overlooked;

(2) 'planning_steps': a list of detailed steps required to trace the reasoning path with. Each step should be a string instead of a dict.

(3) 'declarative_statement': a string of declarative statement that can be transformed from the given query, For example, convert the question 'What do Jamaican people speak?' into the statement 'Jamaican people speak *placeholder*.' leave the *placeholder* unchanged; Ensure the JSON object clearly separates these components.

`In-Context Few-shot`

Q: {Query}

A:

### D.2   DEDUCTIVE-VERIFICATION

You are asked to verify whether the reasoning step follows deductively from the question and the current reasoning path in a deductive manner. If yes return yes, if no, return no".

`In-Context Few-shot`

Whether the conclusion '{declarative_statement}' can be deduced from '{parsed_reasoning_path}', if yes, return yes, if no, return no.

A:

### D.3   ADEQUACY-VERIFICATION

You are asked to verify whether it's sufficient for you to answer the question with the following reasoning path. For each reasoning path, respond with 'Yes' if it is sufficient, and 'No' if it is not. Your response should be either 'Yes' or 'No'.

`In-Context Few-shot`

Whether the reasoning path '{reasoning_path}' be sufficient to answer the query '{Query}', if yes, return yes, if no, return no.

A:

### D.4   BEAM SEARCH

Given a question and the starting entity from a knowledge graph, you are asked to retrieve reasoning paths from the given reasoning paths that are useful for answering the question.

`In-Context Few-shot`

Considering the planning context {plan_context} and the given question {Query}, you are asked to choose the best {beam_width} reasoning paths from the following candidates with the highest probability to lead to a useful reasoning path for answering the question. {reasoning_paths}. Only return the index of the {beam_width} selected reasoning paths in a list.

A:

## D.5 REASONING

Given a question and the associated retrieved reasoning path from a knowledge graph, you are asked to answer the following question based on the reasoning path and your knowledge. Only return the answer to the question.

`In-Context Few-shot`

Question: {Query}

Reasoning path: {reasoning_path}

Only return the answer to the question.

A:

## D.6 DEMONSTRATION OF DEDUCTIVE VERIFICATION

---

**Deductive Verification Example**

**Question:** Who is the ex-wife of Justin Bieber's father?

After one round of beam searching, the **current reasoning path** is:
*Justin_bieber → people.person.father → Jeremy_bieber.*

The **next step candidates** are:
1. *people.married_to.person → Erin Wagner*
2. *people.person.place_of_birth → US, . . .*

The deductive reasoning can be formulated as follows:

**Premises:**

- Justin_bieber → people.person.father → Jeremy_bieber
(from the current reasoning path)
- Jeremy_bieber → people.married_to.person → Erin Wagner
(from the next step candidates)

**Conclusion:**

Erin Wagner is the ex-wife of Justin Bieber's father.
(Using a large language model (LLM) zero-shot approach to reformat the question into a cloze filling task, we use the last entity from the next step candidates, "Erin Wagner", to fill the cloze.)

The prompt will ask whether the conclusion can be deduced from the given premises. If the answer is "yes", return "yes", otherwise return "no."

---

---

**Algorithm 1** Path-RAG Initialization and Retrieval Process

---

1: **Initialization:**
2: **for all** $e_i \in \mathcal{E}, r_i \in \mathcal{R}$ **do**
3:     $z_e^i = \text{LM}(e_i)$                                                        ▷ Embed entities
4:     $z_r^i = \text{LM}(r_i)$                                                      ▷ Embed relations
5: **end for**
6: Populate nearest neighbor index with $\{z_e^i\}$ and $\{z_r^i\}$     ▷ Facilitate retrieval
7: **procedure** RETRIEVE(query $q$)
8:     $\mathcal{K}_i = \text{LM}(\text{'prompt'}, q)$                        ▷ Generate keywords
9:     **for all** $k_i^m \in \mathcal{K}_i$ **do**
10:         $k_i \leftarrow \text{concatenate}(k_i^m)$
11:         $z_k = \text{LM}(k_i)$                ▷ Embed concatenated keywords
12:         $\mathcal{E}_k = \text{argtopk}_{i \in \mathcal{E}} \cos(z_k, z_e^i)$     ▷ Retrieve top-k entities
13:         $\mathcal{R}_k = \text{argtopk}_{i \in \mathcal{R}} \cos(z_k, z_r^i)$     ▷ Retrieve top-k relations
14:     **end for**
15:     **return** $\mathcal{E}_k, \mathcal{R}_k$
16: **end procedure**
17: **procedure** SCOREPATH($\mathcal{E}_k, \mathcal{R}_k$)
18:     Initialize Score $\leftarrow 0$
19:     **for** each $e_k \in \mathcal{E}_k$ and $r_k \in \mathcal{R}_k$ **do**
20:         Calculate $S_e^i, S_r^i \leftarrow \cos(z_k, z_e^i), \cos(z_k, z_r^i)$   ▷ Compute similarity scores
21:         $S(p) = S_r^i + S_e^i + \alpha \max_{\forall j \in N_i}(S_r^j + S_e^j)$   ▷ Score path using Eq. 2
22:         Score $\leftarrow \max(\text{Score}, S(p))$                   ▷ Update max score
23:     **end for**
24:     **return** Score, $p$
25: **end procedure**

---

**Algorithm 2** Deductive-Verification Guided Beam Search

---

**Require:** User query $x$, Beam width $B$
**Ensure:** Reasoning path $s^{1:T}$
1: Initialize $\mathcal{H}_0 = \{\emptyset\}$
2: Utilize LLM to generate from $x$:
3:     Planning steps.
4:     Declarative statement $x'$.
5: **for** $t = 1$ to $T$ **do**
6:     **for** each $h \in \mathcal{H}_{t-1}$ **do**
7:         Generate possible next steps $s^t \in \mathcal{S}$ using Path-RAG.
8:         **for** each $s^t$ **do**
9:             Compute $C(x', s^t, s^{1:t-1})$ using LLM:
10: 
$$C(x', s^t, s^{1:t-1}) = \begin{cases} 1 & \text{if } x' \text{ can be deduced from } s^t \text{ and } s^{1:t-1}, \\ 0 & \text{otherwise.} \end{cases}$$

11:             **if** $C(x', s^t, s^{1:t-1}) = 1$ **then**
12:                 Append $s^t$ to $h$ to form new hypothesis $h'$.
13:                 Add $h'$ to $\mathcal{H}_t$.
14:             **end if**
15:         **end for**
16:     **end for**
17:     $\mathcal{H}_t = \text{Top}_B(\mathcal{H}_t)$ based on scoring function (like plausibility or likelihood).
18: **end for**
19: **return** the best hypothesis from $\mathcal{H}_T$.

---

