# OpenReview forum: "FiDeLiS: Faithful Reasoning in Large Language Models for Knowledge Graph Question Answering"
_ICLR.cc/2025/Workshop/BuildingTrust — BuildingTrust_

### Official Review · Reviewer_MKR3 · 2025-03-02
**Well-written paper with overall and stepwise evaluations**

**Rating:** 8
**Confidence:** 4

**Review:**

**Summary**
This paper is addressing the hallucination problem of LLM reasoning by retrieving relative knowledge from knowledge graphs (KG). The paper proposed two modules: PATH-REG and DVBS, which aims to retrieve the relevant and accurate information from the KG efficiently. The experiments verifies that their proposed method can enhance the reasoning ability of LLMs, and the ablation studies verifies each of their design choice.

**Strengths**
1. The paper is well-written, very clear, and easy to understand and follow the ideas.
2. The experiments evaluates the model on the both the final and stepwise reasoning performance, which confirms advantages of the proposed method.

**Weakness**
1. In Line 103, in the relationship Justin Bieber ---> Jeremy Bieber --> Erin Wagner, is the subject of two relationships are Jeremy Bieber? If not, should the second relationship be ex-husband? I am not familiar with KGs, so I am not very sure about this.
2. In Section 3.1 Line 199-200, the paper computing the score $S_0((r_j, e_j))$ by only examining entities within one-hop. Is it possible to extend this to multi-hop, since my intuition is that probably multi-hop ($H\geq2$) can improve the accuracy of the retrieval, but is less efficient which requires more computational resources. So I suggest authors add more discussion about this.
3. How is the performance of using LLMs to do the deductive verification? Is there any potential limitations using this LLM-as-a-Judge methodology, given the fact it has position bias, length bias, etc.


**Sidenote**
1. I suggest authors read about the PLASMA [1] paper, which uses the similar idea of verification-based beam search as this paper but on planning tasks.
[1] Brahman, Faeze, et al. "Plasma: Making small language models better procedural knowledge models for (counterfactual) planning." arXiv preprint arXiv:2305.19472 (2023).

---

### Official Review · Reviewer_UJ1d · 2025-03-02
**FiDeLiS addresses a critical challenge in LLM reasoning—namely, the prevalence of hallucinated or factually incorrect outputs—by leveraging external structured knowledge from KGs. While the paper is motivated by an important problem and proposes an intriguing two-pronged approach, it suffers from several significant shortcomings that undermine its overall contribution.**

**Rating:** 4
**Confidence:** 4

**Review:**

Strengths:

1.Integrated Framework:
 By combining a retrieval-augmented module (Path-RAG) with a deductive verification beam search (DVBS), the method aims to ensure that every reasoning step is both semantically relevant and logically valid.

2.Empirical Evaluation:
 Extensive experiments on KGQA benchmarks (such as WebQSP, CWQ, and others) demonstrate that FiDeLiS can outperform several strong baselines, including methods like ToG and RoG.

Weaknesses:

1.Computational Inefficiency:
 Despite claims of efficiency gains via candidate pre-selection, the heavy reliance on step-wise beam search and LLM interactions results in high latency. The trade-off between computational cost and improved reasoning fidelity is not sufficiently justified.

2.Questionable Novelty:
 FiDeLiS largely builds on existing retrieval-augmented frameworks and deductive verification techniques. The integration of these components, though non-trivial, does not represent a significant leap in innovation.

3.Reliability Issues:
 Error analysis reveals that a substantial portion of the generated reasoning steps (only around 67% validity) are either formatted incorrectly or reference non-existent KG facts. This undermines the paper’s central claim of achieving faithful reasoning.

4.Organization and clarity:
 Critical details—such as the rationale behind key design choices and hyperparameter settings—are not clearly articulated

---

### Official Review · Reviewer_J21u · 2025-03-02
**Good Paper for Ensuring Fidelity of Knowledge-Based Reasoning**

**Rating:** 6
**Confidence:** 4

**Review:**

This paper tackles the important problem of ensuring faithfulness to retrieved knowledge graph entries in multi-hop reasoning tasks. They identify two key challenges faced by such methods. Firstly, retrieval systems may be insufficient to collect all relevant knowledge graph edges simply given the initial query. Secondly, the generated reasoning chains may have hallucinations, fail to terminate, or terminate inappropriately early. Both failure modes introduced are interesting and represent important problems for the deployment of knowledge-graph based reasoning in LLMs. Then, they propose two methods for improving these issues (1) a Path-Rag system which takes into account the structural connectivity of knowledge graph and (2) a deductive beam search strategy which efficiently explores reasoning chains and ensures they obey local and global coherence constraints. They show the superior performance of this method relative to standard prompting, chain of thought, and RAG. Overall, I like this paper. The proposed modifications are well-motivated and justified by the provided ablations. Moreover, this paper is generally well-written and easy to follow. In terms of questions, I wonder why the authors chose to not validate any of the new test-time reasoning models with and without their framework (i.e. o1 and r1). I think the models evaluated currently are a bit behind the state-of-the-art. Moreover, I was slightly confused about whether the "retrieval" is run only once in the beginning or to generate candidate relations after every step of reasoning. In the first case, the evaluation of different relations could still suffer from being too "local" to the initial query, whereas the second could result in some efficiency issues. It would be quite interesting if the authors could discuss this more explicitly. Lastly, I wonder how the number of retrieved knowledge graph entries (denote by M) should be set. Could it vary based on the query? Overall, however, I think this is an interesting and valuable contribution and would like it to be accepted.

---

### Decision · Program_Chairs · 2025-03-04

Accept